# Gene Expression in MC3T3-E1 Cells Treated with Diclofenac and Methylprednisolone

**DOI:** 10.3390/genes14010184

**Published:** 2023-01-10

**Authors:** Tomasz P. Lehmann, Ewa Iwańczyk-Skalska, Jerzy Harasymczuk, Paweł P. Jagodziński, Maciej Głowacki

**Affiliations:** 1Department of Biochemistry and Molecular Biology, Poznań University of Medical Sciences, Święcickiego 6, 60-781 Poznan, Poland; 2Department of Paediatric Surgery, Traumatology and Urology, Poznan University of Medical Sciences, Szpitalna 27/33, 60-572 Poznan, Poland; 3Department of Paediatric Orthopaedics and Traumatology, Poznan University of Medical Sciences, 28 Czerwca 1956 135/147, 61-545 Poznan, Poland

**Keywords:** MC3T3, diclofenac, methylprednisolone, glucocorticoid, nonsteroidal anti-inflammatory drugs

## Abstract

Nonsteroidal anti-inflammatory drugs (NSAIDs) and glucocorticoids (GCs) are often used to treat articular-skeletal disorders. The extended use of NSAIDs and GCs have adverse effects on bone metabolism, reducing bone quality and impairing fracture healing. In the present study, we used mouse pre-osteoblast cells MC3T3-E1 to demonstrate the effects of diclofenac (DF) and methylprednisolone (MP) on cell proliferation and gene expression. Cells were incubated with three doses of DF or MP: 0.5 µM, 5 µM, and 50 µM. MP decreased cell viability even after 24 h, but DF inhibited cell viability after only seven days of treatment. The cells were lysed after one, two, three, and seven days of treatment, and gene expression was analyzed by reverse transcription and quantitative PCR (RT-qPCR) assays. DF did not significantly affect the expression of the osteogenic marker genes. MP modified the expression of *Osx*, *Runx*, and *Col1a1*. We concluded that MP is a more potent inhibitor of mouse pre-osteoblast differentiation and viability than is DF. Our results suggest that prolonged DF treatment could be less harmful to osteoblasts than MP treatment.

## 1. Introduction

Nonsteroidal anti-inflammatory drugs (NSAIDs) and glucocorticoids (GCs) are frequently used in addition to opioids to control pain. NSAIDs are one of the most used therapeutics which may be obtained over the counter in many countries. NSAIDs are used for a wide variety of indications, including short-term and long-term treatment of pain [1]. Long-term pain accompanies inflammation in patients with chronic musculoskeletal diseases, fractures, and osteoarthritis [2]. Diclofenac (DF) and methylprednisolone (MP) are prescribed after surgery for postoperative pain management. Long-term and even low-dose glucocorticoid treatment can cause decreased osteoblast viability and function, leading to osteoporosis and osteonecrosis [3]. The chronic administration of GCs exhibits a substantial risk factor for fragility and osteopenia in humans and mice. Numerous studies have demonstrated that GCs negatively affect bone metabolism, leading to the deterioration of tissue quantity and quality [4,5]. GCs act directly on the glucocorticoid receptors in osteoblasts and osteocytes, inducing apoptosis, but they are also essential in regulating genes involved in osteoblast differentiation [3].

In the human body, under the influence of drugs, as well as local processes (metabolic disorders, genetic disease), the bone structure may be weakened in some situations, leading to pathological titers [6,7,8].

In combination with GCs in chronic articular-skeletal pain treatment, DF is frequently prescribed. Diclofenac, sodium 2-(2,6-dichloroanilino)-phenylacetate, is a member of the aryl acetic acid group of NSAIDs [9]. The anti-inflammatory efficacy of DF is undeniable; however, the side effect on bone healing has been reported [10]. The chronic administration of NSAIDs, such as DF, negatively affects the skeletal system [11]. Earlier results have suggested that DF mediates cyclooxygenase 2 (COX-2) inhibition in the early phase of bone healing and harms osteoblast, osteoclast, and macrophage activity [12].

Many studies have shown that NSAIDs, including DF, decreased osteoblast proliferation and differentiation [13,14,15]. The typical mechanism of NSAIDs and DF action on cells is the inhibition of cyclooxygenases and subsequently, prostaglandin synthesis [16]. DF and other NSAIDs can interfere with the human and mouse bone marrow mesenchymal stem cell cycle by arresting the G0/G1 phase [17]. In the human MG-63 osteosarcoma cell line, however, treatment with DF, which reduces cell proliferation capacity, produced no cell cycle changes [18]. DF also inhibited osteoclast differentiation, without triggering cell apoptosis [19]. DF induced the accumulation of the inhibitor of kappa B in the cytosol, which led to the suppression of the nuclear translocation of NFκB (nuclear factor κ-light-chain-enhancer of activated B cells). There have been several types of research regarding the impact of DF on mouse MC3T3-E1 cells. These pre-osteoblasts can synthesize prostaglandin; nevertheless, NSAIDs diminish MC3T3-E1 proliferation by cyclooxygenase and prostaglandin-independent mechanisms [9,11].

The adverse effects of NSAIDs and GCs on bone are well known, but the DF mechanism and its impact on gene expression have yet to be well established. Some gene expression studies have already been conducted on MC3T3-E1. Hadjicharalambous states that neither DF nor any of the NSAIDs investigated in their study inhibit *Runx2* or *Bsp* gene expression [14]. In the D1-cell mouse mesenchymal stem cell line, DF showed no significant effect on expressions of the osteogenic markers, *Runx2*, *Bmp2*, *Col1a1*, and *Ocn* (osteocalcin) [15]. Melguizo-Rodríguez et al. found that in human osteoblast lines, DF and other NSAIDs inhibited the expression of *RUNX2*, *COL1A1*, *OSX*, and *BGLAP* (osteocalcin) [20]. The sensitivity of gene expression to the presence of DF and NSAIDs depends on the cell type.

A short review of the literature concerning NSAIDs’ effect on bone has revealed the ambiguity of the presented data in human, animal, and in vitro studies [21]. The mechanisms which cause this ambiguity need to be better understood. To date, there is no consensus regarding the impact of DF on bone healing and the effects of DF on cultured cells. Therefore, we looked for a cellular model which is closely related to animal species. Previously using the C57BL/6J mouse strain, we demonstrated that DF decreased murine bone mineral density [22]. To explain this decrease in bone mineral density, we hypothesized osteoblast dysfunction caused by DF. As the experimental model, we used MC3T3-E1 pre-osteoblast cells derived from the C57BL/6J mouse strain.

In the present study, we used the mouse osteoblast progenitor cells MC3T3-E1 to demonstrate DF and MP treatment on cell proliferation and gene expression. Osteoblasts participate in bone formation and remodeling, promoting the regeneration of damaged bone tissue [12]. Therefore, using MC3T3-E1 cells, we addressed the objective of this study. The study will improve our understanding of the role of osteoblasts in DF mechanisms on bone turnover. This in vitro experimental study may contribute to elucidating the mechanism underlying the action of NSAIDs on osteoblasts and, therefore, on bone.

## 2. Materials and Methods

### 2.1. Cell Culture

The MC3T3-E1 subclone 4 pre-osteoblast, mouse (*Mus musculus*) -derived cell line was purchased from ATCC-CRL-2593. Minimum Essential Eagle’s Medium (α-MEM) (Corning, Corning, NY, USA), ascorbic acid (Avantor, Gliwice, Poland), β-glycerophosphate (Cayman Chemical, Ann Arbor, MI, USA) 100 units penicillin, 100 µg streptomycin, 250 ng amphotericin B per mL (ABAM, Sigma, St. Louis, MO, USA), fetal bovine serum (FBS, Biowest, Nuaillé, France), and trypsin/EDTA (Biowest) were also obtained [23]. We also used proliferation medium: α-MEM, 10% FBS, 1%vol. ABAM; differentiation medium: α-MEM, 10% FBS, 1%vol. ABAM, β-glycerophosphate 10 mM, ascorbic acid 50 µg/mL. Diclofenac (DF, J62609, Alfa Aesar, Kandel Germany, diclofenac sodium salt CAS number 15307-79-650) 50 mM stock in DMSO (dimethyl sulfoxide, sterile, catalogue number, DMS666, CAS Number 67-68-5 BioShop Canada Inc., Mainway Burlington, Ontario, Canada), final concentrations in differentiation culture media 50, 5, and 0.5 µM, methylprednisolone (MP, PHR1717-500MG, Merck, Darmstadt, Germany, C.A.S. Number 83-43-2) 50 mM stock in DMSO, final concentrations in differentiation culture media 50, 5 and 0.5 µM were also used.

### 2.2. Cell Viability and Caspase Activity

The 3-(4,5-dimethylthi-azol-2-yl)-2,5-diphenyltetrazolium bromide (MTT)-based Cell Growth Determination kit (Sigma) was used for cell viability analyses. Cells were incubated for 24, 48, 72, or 168 h with test substances in 24-well plates in the differentiation medium: α-MEM, 10% FBS, 1% vol. ABAM, β-glycerophosphate 10 mM, ascorbic acid 50 µg/mL. Finally, cells were incubated for 4 h in a differentiation medium with the addition of MTT (0.5 mg/mL). Subsequently, the cell culture medium was removed, and the solvent was added (1 mM hydrochloric acid in isopropanol anhydride (Avantor)). Following gentle mixing, analysis was performed using the Stat-Fax 2100 spectrophotometer (Awareness Technology Inc., Palm City, FL, USA) at a wavelength of 630 nm (background absorbance was measured at 405 nm).

Caspase activity was analyzed by a Caspase-Glo 3/7 assay (Promega Corporation, Madison, WI, USA). Following cell stimulation, Caspase-Glo 3/7 Reagent was added (culture medium: Caspase Glo 3/7 reagent, 4:1) to cells in a 96-well plate. After incubation for 1 h, luminescence was measured in the TD20/20 luminometer (Turner Designs Inc., Sunnyvale, CA, USA).

### 2.3. RNA Extraction and RT-PCR

Total RNA was extracted from the cultured MC3T3-E1 cells using TRItidy G, according to the manufacturer’s instructions (A4051,0100, Applichem, Darmstadt, Germany), and the High Pure Viral RNA Kit (1185882001, Roche, Mannheim, Germany) was used to purify the RNA. One µg of total RNA was used to synthesize cDNA in a 20 µL reaction containing 2.5–5 u/µL M-MLV Reverse Transcriptase Kit (28025-013, Thermo Fisher Scientific, Waltham, MA, USA), 1 u/µL RNaseOUT™ Recombinant Ribonuclease Inhibitor (10777-019, Thermo Fisher Scientific), 25 ng/µL oligo(dT)_24_, 10 ng/µL random hexamers, 2 mM dNTPs, and 10 mM DTT. The temperature program was: 65 °C for 5 min, ice chilling, 37 °C for 50 min, and 70 °C for 15 min.

### 2.4. Gene Expression

Gene expression was evaluated by qPCR, using the assays listed in Table 1. TaqMan Fast Advance Mix kit (4444963, Thermo Fisher Scientific) and LightCycler 96 or 480 (Roche Diagnostics, Basel, Switzerland) were used. The reaction conditions were the same for all assays: preheating 50 °C for 120 s, 95 °C for 120 s; amplification at 95 °C for 3 s, 60 °C for 30 s, repeated 45 times. The reaction mixture was: TaqMan Fast Advanced Master mix (2×) 5 µL, TaqMan Assay (20×) 0.5 µL, water 3.5 µL, and cDNA 1 µL. The crossing-point values were calculated automatically based on the second derivative algorithm, and the results were analyzed by the relative expression method, as in our previous work [24,25,26]. *Hprt* and *B2m* were used as reference genes.

### 2.5. Statistical Analysis

The data were analyzed using GraphPad InStat version 7.00 software (GraphPad Software, San Diego, CA, USA) and Statistica 13.0 software (TIBCO Software, Inc., Palo Alto, CA, USA). One-way ANOVA with Tukey’s post hoc test was used. The significance level was set at 5% (*p* < 0.05).

Real-time data were analyzed using Statistica and REST, the expression analysis software tool [25]. At first, the crossing-point values were calculated automatically based on the second derivative algorithm using the software LightCyclr 96 1.1.0.1320 (Roche Diagnostics International). The results were calculated to relative expression values using a standard curve specific for each gene. The relative values obtained were divided by the geometric mean of relative values of the reference genes *Hprt* and *B2m*. The geometric means of four independent biological repeats were used to calculate the results. The standard error of the mean was calculated using four biological repeat replications. The control expression levels at 24 h of incubation, without substances, were assumed to be one. Statistical analysis was performed on the logarithms of the relative expression values to obtain additive values, normality was confirmed using the Schapiro–Wilk’s test, and Leven’s test for homogeneity was performed, as well as the Tukey post-hoc test using Statistica 13.0 software (TIBCO Software, Inc.). Alternatively, the expression results were analyzed using the REST protocol [27]. These analyses results are available as Appendix A.

## 3. Results

### 3.1. Cell Viability

MC3T3-E1 cells were treated with DF and MP to determine whether these substances induced changes in cell viability. In Figure 1, we show that following 24 h of treatment with 50 µM MP, the MC3T3-E1 cell viability was decreased by 20%. The same treatment time with 50 µM DF did not reduce the viability of the MC3T3-E1 cells. The treatment of MC3T3-E1 cells with 0.5, 5, and 50 µM MP for 48 h diminished cell viability by almost 20%. The treatment of MC3T3-E1 cells with 0.5, 5, and 50 µM MP for 72 h diminished cell viability by 25–30%.

Treatment of MC3T3-E1 cells with 0.5 µM, 5 µM, and 50 µM MP for seven days diminished cell viability by more than 40%. The seven-day treatment time with 50 µM DF decreased the viability of MC3T3-E1 cells by almost 20%. The substances were administered once for each course experiment with the new differentiation medium. Over the seven days, the cells retreated after three days with the medium replacement procedure.

### 3.2. Caspase

MC3T3-E1 cells were treated with DF and MP to determine whether these drugs induced changes in the caspase-3 and -7 activities. We used the Caspase-Glo 3/7 assay to achieve the goal, which provides a proluminescent caspase-3/7 DVD-aminoluciferin substrate and a luciferase in reagent optimized for caspase-3/7 activities. Figure 2 shows that following 72 h of treatment with MP 0.5, 5, and 50 µM, the caspase-3 activity decreased in the MC3T3-E1 cells by 50%. The same treatment time with DF did not affect caspase 3 in the MC3T3-E1 cells. The treatment of MC3T3-E1 cells with MP 0.5, 5, and 50 µM for seven days diminished cell viability by 40%. Due to the diminished caspase-3 activity caused by MP, we determined that caspase-3 was not involved in the mechanism responsible for the decreased viability of MC3T3-E1 cells or their apoptosis. DF was not involved in the regulation of caspase-3 activations.

### 3.3. Gene Expression

Runx2 and Dlx5 homeobox transcription factors activate the expression of osterix (*Osx*, *Sp7*), which, in turn, activates the expression of the genes Satb2, encoding a transcription factor, and *Col1a1*, encoding collagen I [28]. Runx2, Dlx5 and Osx are key transcription factors associated with the differentiation of mesenchymal precursor cells into osteoblasts and osteocytes [29,30]. Figure 3 shows the relative expression of *Osx* (A and B,), *Runx2* (C and D), *Opn* (E and F), and *Col1a1* (G and H). Figure 4 shows *Tgfb1* (A and B), *Alp* (C and D), *Satb2* (E and F) and *Dlx5* (G and H) values obtained by RT-PCR with TaqMan probes. *Hprt* and *B2m* were used as references for the mRNA level. DF did not significantly regulate gene expression. MP increased the expression of *Runx2* after 72 h (Figure 3F). MP also increased the expression of *Opn* after seven days of treatment (Figure 3D). The expression of *Col1a1* was diminished after seven days (Figure 3H).

## 4. Discussion

It has been noted that several non-selective NSAIDs and COX-2 selective inhibitors suppress bone remodeling. Bone repair has also been diminished in vivo as a side effect of pain treatment [9]. Depending on the study protocol, chronic administration of NSAIDs adversely affects the skeletal system and bone healing, and may also have a neutral impact [21]. Our previous study demonstrated that COX-2 inhibitor DF decreased bone mineral density in C57BL/6J mice [22]. Our present study shows that DF did not affect the gene expression of the osteogenic markers we tested. We revealed that DF exhibited decreased MC3T3-E1 viability after seven days, but the deregulation of the marker genes did not accompany this effect. MP effectively reduced cell viability, decreasing *Col1a1* and increasing *Runx2* expression.

MC3T3-E1 osteoblast progenitor cells originate from the C57BL/6 mouse strain. A suitable model to investigate NSAIDs’ effect on osteoblasts is isolated from mouse calvaria. MC3T3-E1 cells, by COX-2 induction, produce prostaglandins which trigger the prostaglandin extracellular receptor and the cAMP pathway [31]. Prostaglandin E could induce the proliferation of MC3T3-El and stimulated Egr-1, c-fos and c-jun expressions in MC3T3-E1 cells [32,33,34]. Prostaglandin is produced by MC3T3-E1 and stimulates its mineralization [35]. Since it has been revealed that MC3T3-E1 has secreted prostaglandin E and initiated the cAMP pathway, DF activity on MC3T3-E1 should reverse this effect.

DF is among the most effective NSAIDs, and it is a potent inhibitor of COX-2 and COX-1 [16]. Even though conclusive evidence regarding the teratogenicity of diclofenac in human embryos is lacking, animal studies have shown that administration of the drug at a high concentration in early gestation inhibits implantation and embryonic development of the bones [36]. It crosses the human placenta readily during the first trimester and should be considered as potentially teratogenic [37]. Previous in vivo and in vitro studies showed that DF decreased bone healing and decreased the viability of preosteoclasts [14]. DF is a type of NSAID that inhibits prostaglandin synthesis and affects bone cell osteogenic differentiation in cultured osteoblasts [9,14]. The exact mechanism of DF action in MC3T3-E1 cells, especially its effect on gene expression, remains unclear. Contrary to DF, methyloprednisolone, which easily enters into cells, is one of the GCs that strongly inhibits bone mineralization and differentiation.

As we expected, MP decreased cell viability. This observed decrease in cell viability cannot be explained by apoptosis because apoptotic marker caspase-3/7 was not increased. Eight marker genes were analyzed to reveal the effect of MP on pre-osteoblasts. In our study, after 72 h, *Runx2* and *Opn* expression increased, and *Col1a1* decreased in the presence of MP. However, in the earliest studies, in MC3T3-E1 cultures, it has been shown that GCs, either at physiological or pharmacological concentrations (DEX 1 µM for 72 hr), induce only modest alterations in the levels of *Runx2* type I and type II [38]. We confirmed a strong MP effect diminishing osteoblast proliferation and decreasing gene expression. One possible explanation for these unexpected results regarding *Runx2* and *Opn* gene expression in our study could be the inconsistencies between the gene expression profile of MC3T3-E1 subclone four and the primary calvarial osteoblasts [39]. The subclone information is not always available in publications [40]. These results partially confirm the earlier reported effect of MP on osteoblast cells and gene expression. Several authors have shown that DF has a weak impact on pre-osteoblast viability compared to MP.

It has been shown in previous studies that NSAIDs, including DF, decreased the MC3T3-E1 viability [14]. Our data are consistent with former observations, showing only a slight 20% decrease in MC3T3-E1 viability after seven days of DF treatment. The mechanisms by which NSAIDs exert their effect on pre-osteoblast cell growth are still under debate. In addition to the COX-2 dependent mechanism, DF could exert an antiproliferative effect using several alternative pathways [16]. In rat calvaria osteoprogenitor cells, therapeutic doses of DF promoted cell death by arresting their cell cycle in phase G0/G1, which may suppress tissue formation and bone remodeling [9].

To provide more insight into the mechanism of the action of DF in the pre-osteoblastic MC3T3-E1 cells, we investigated osteogenic marker gene expression. We did not observe significant variation in gene expression in the presence of DF. Hadjicharalambous et al. applied 10^−5^ and 10^−6^ DF in MC3T3-E1 cells for 5, 10, and 15 days, but the authors did not investigate several prominent osteogenic marker genes in the treated cells [14]. Hadjicharalambous et al., using MC3T3-E1 incubated with several COX-2 selective NSAIDs, revealed that none of the NSAIDs affected the expression of either the *Runx2* or *Bsp* genes. In human primary osteoblast cell cultures, *RUNX-2*, *COL1A1*, and *OSX* expression were reduced by treatment with all NSAIDs and DF [20].

We checked DF in pre-osteoblast cells to reveal how DF decreased bone mineral density in the C57BL/6J mice. Although in our previous study, the mouse cells were treated with DF for 28 days, the short-term gene expression assay in the MC3T3-E1 cells ought to reveal at least minor changes in gene expression. Since the gene expression in MC3T3-E1 was not significantly modified by DF, the explanation for the decrease in bone mineral density in mice is likely due to other mechanisms. The effect might be due to DF’s impact on osteoclasts or mature osteoblasts, or possibly the systemic effect of DF on steroid hormone secretion in the gonads.

The limitation of the study is that there was not an application of another cell line as an osteoclast to explain the effects of DF on bone mineral density. Since the mice in our former experiment received DF daily by injection, the cell culture may also require daily retreatment with DF and MP. The mechanism of MP action on *Col1a1*, *Runx2*, and *Opn* gene expression and their interplay should be further investigated. The MP action on cell proliferation, independent of prostaglandin, should be exploited. Future work in deciphering NSAIDs and GCs should answer the question regarding the mechanism of MP action involving or not involving prostaglandin synthesis. The effect of DF on MC3T3-E1 should be verified concerning the *COX-1* and *COX-2* expression level and prostaglandin production.

## 5. Conclusions

We concluded that DF exhibits fewer harmful effects on the osteoblasts than does MP. For the first time, we showed that the previoulsy observed decrease in MC3T3-E1 proliferation is not strongly associated with gene expression in pre-osteoblasts.

## Figures and Tables

**Figure 1 genes-14-00184-f001:**
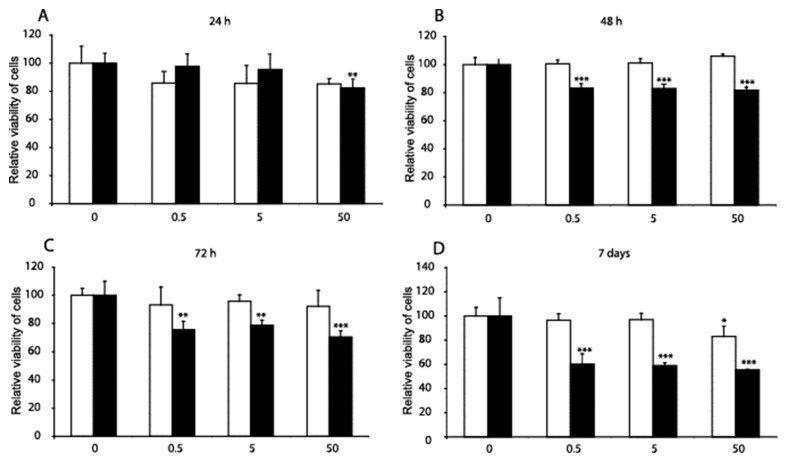
**Comparison of the effects of diclofenac (DF) and methylprednisolone (MP) on the viability of MC3T3-E1 pre-osteoblast cells.** The MC3T3-E1 cells were treated for 24 h, 48 h, 72 h, and 168 h (**A**–**D**) with three concentrations of DF (open bars) and MP (black bars): 0.5 µM, 5 µM, and 50 µM. The viability of the cells, determined by the 3-(4,5-dimethylthi- azol-2-yl)-2,5-diphenyltetrazolium bromide (MTT) assay, was expressed as the mean ± SD from four experiments and was compared to the untreated control. It was assumed that the viability of the control cells equals 100%. The significance was evaluated using one-way ANOVA with Tukey’s post hoc test. The significance level was set at 5% (*p* < 0.05); significant at: * *p* < 0.05, ** *p* < 0.01, and *** *p* < 0.001.

**Figure 2 genes-14-00184-f002:**
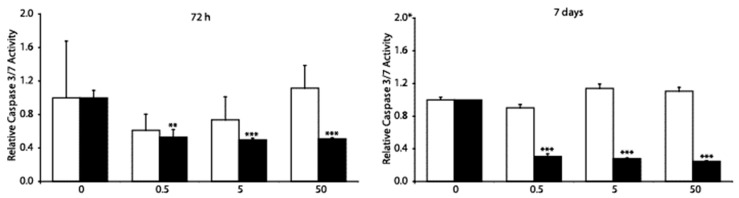
**Comparison of the impact of MP and DF on the caspase-3 and caspase-7 activities in MC3T3-E1 pre-osteoblast cells**. After 72 h and seven days of treatment with DF (open bars) and MP (black bars), the MC3T3-E1 cells were lysed. The activity of caspase-3 and caspase-7 were determined in triplicate using the Caspase-Glo 3/7 assay (Promega Corporation, Madison, WI, USA), as described in the Materials and Methods section. Results were the mean ± SD of four separate experiments, the untreated control assuming that the caspase activity of the control cells equals 1. The significance was evaluated using one-way ANOVA with Tukey’s post hoc test. The significance level was set at 5% (* *p* < 0.05); significant at: ** *p* < 0.01, and *** *p* < 0.001.

**Figure 3 genes-14-00184-f003:**
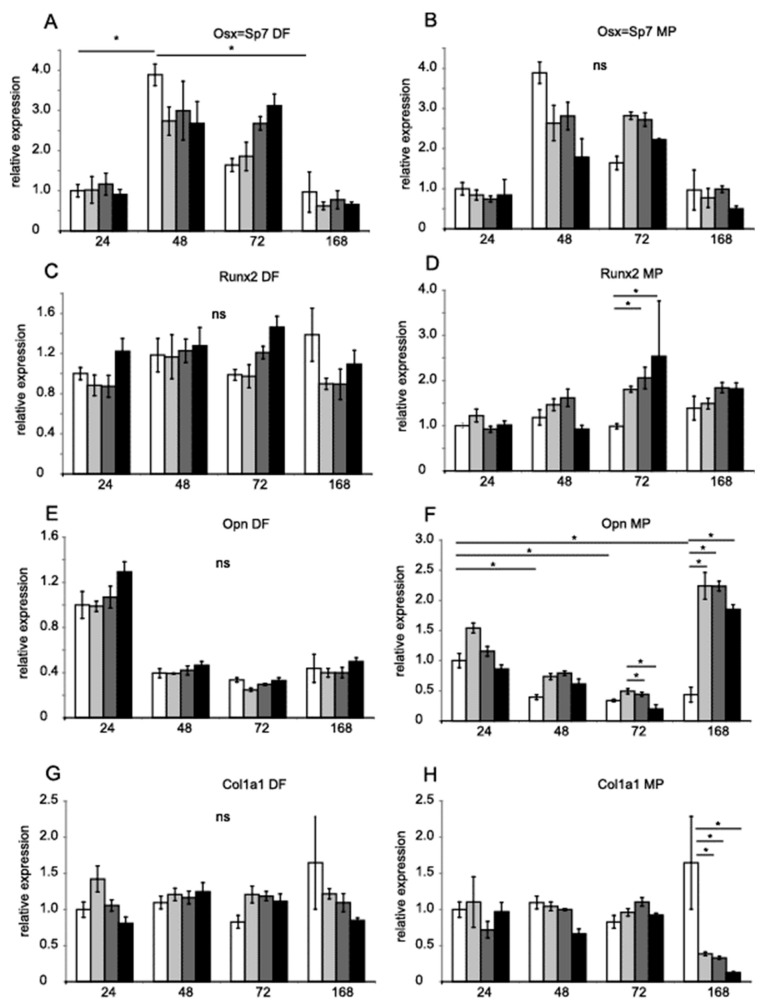
**Gene expression in MC3T3-E1 in the presence of DF and MP**. *Osx*, *Runx2*, *Opn*, and *Col1a1* mRNA levels in MC3T3-E1 cells treated with DF (**A**,**C**,**E**,**G**) 0.5 µM, 5 µM, 50 µM (increasing grey scale, respectively) for 24 h, 48 h, 72 h, and 168 h. MC3T3-E1 cells were treated with MP (**B**,**D**,**F**,**H**) 0.5, 5, 50 µM (increasing grey scale, respectively) for 24 h, 48 h, 72 h, and 168 h. Untreated control equals 1 (open bars). Results are presented as geometric means of relative expression. *Hprt* and *B2m* were used as reference genes. The significance was evaluated using one-way ANOVA with Tukey’s post hoc test. The significance level was set at 5% (*p* < 0.05). Error bars represent ± SEM. * *p* < 0.05.

**Figure 4 genes-14-00184-f004:**
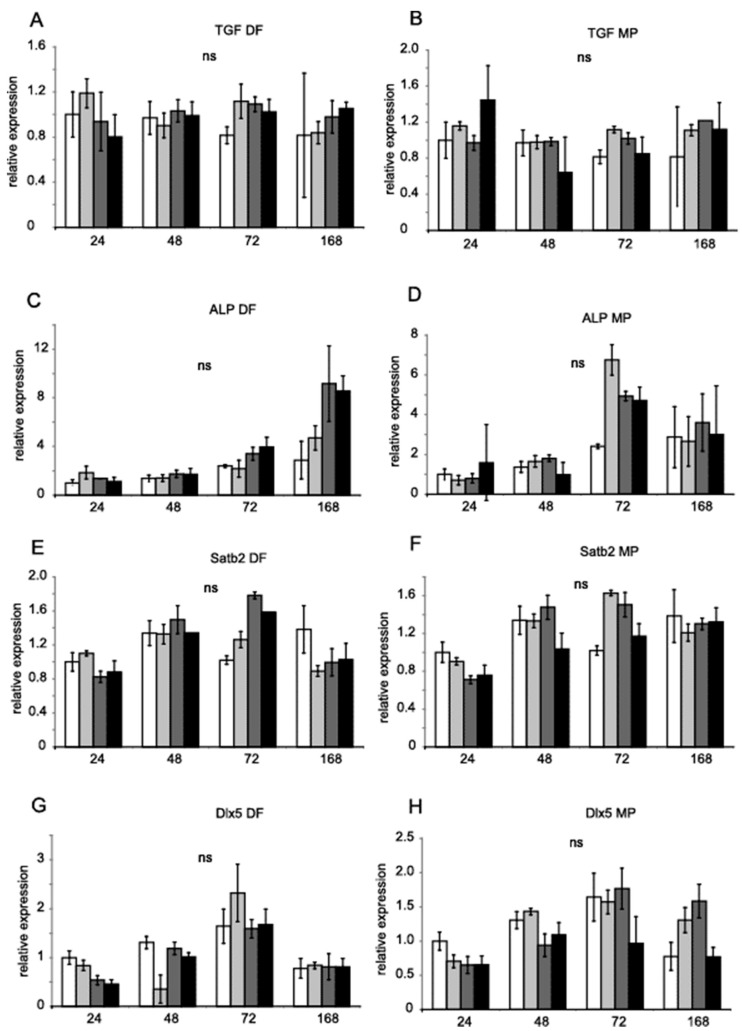
**Gene expression in MC3T3-E1 in the presence of DF and MP**. *Tgfb1*, *Alp*, *Satb2*, and *Dlx5* mRNA levels in MC3T3-E1 cells treated with DF (**A**,**C**,**E**,**G**) 0.5 µM, 5 µM, 50 µM (increasing grey scale, respectively) for 24 h, 48 h, 72 h, and 168 h. MC3T3-E1 cells were treated with MP (**B**,**D**,**F**,**H**) 0.5 µM, 5 µM, 50 µM (increasing grey scale, respectively) for 24 h, 48 h, 72 h, and 168 h. Untreated control equals 1 (open bars). Results are presented as geometric means of relative expression. *Hprt* and *B2m* were used as reference genes. The significance was evaluated using one-way ANOVA with Tukey’s post hoc test. The significance level was set at 5% (*p* < 0.05). Error bars represent ± SEM.

**Table 1 genes-14-00184-t001:** Gene and assay ID numbers.

Gene	Assay Catalogue Number	Sequence ID
*Osx*	4351372	Mm04933803_m1
*Runx2*	4331182	Mm00501584_m1
*Opn/Spp1*	4331182	Mm00436767_m1
*Col1a1*	4331182	Mm00801666_g1
*Tgfb1*	4331182	Mm01178820_m1
*Alp*	4331182	Mm00475834_m1
*Satb2*	4331182	Mm00507331_m1
*Dlx5*	4331182	Mm00438430_m1
*Hprt*	4331182	Mm03024075_m1
*B2m*	4331182	Mm00437764_m1

## Data Availability

Data is contained within the article or Appendix A.

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
