# Peer review of "Gene Expression in MC3T3-E1 Cells Treated with Diclofenac and Methylprednisolone"

_genes, 2023, doi:10.3390/genes14010184_

Round 1
Reviewer 1 Report (Previous Reviewer 1)
The authors have revised the manuscript incorporating most of the possible suggestion in the first round of review in a satisfactory manner. Statistical details were added to Figures 1 to 4. Satisfactory explanation were provided for the missing DMSO controls. Other minor errors ( typos, incorrect gene symbols, concentration. missing references, duplication etc were also taken care of. Recommend the revised manuscript for publication
Author Response
We thank the Reviewer for accepting our efforts in proofreading and editing the text of the manuscript.
Reviewer 2 Report (Previous Reviewer 2)
The authors utilise a mouse strain which is pre-osteoblastic in nature to test the hypothesis that decrease in proliferation upon NSAID/GC treatment is due to effect on the expression of genes involved in osteogenesis. They conclude that at least in this pre-osteogenic strain this is not the case, and DF is a safer drug compared to MP.
Specific comments to improve scientific soundness of the article:
1. Introduction: Clearly mention the effect of DF/MP as known in individuals who are healthy and those with musculoskeletal diseases. Please clarify that these NSAIDs are general pain management drugs.
2. Materials and methods: Was the cell culture for gene expression assays upon DF and MP addition done in presence of antibiotics? If yes, another set of the same experiments should be performed in the absence of antibiotics as antibiotics may have an effect on action of drugs on cell lines.
3. Figure 3, panel E: Please redo statistical analysis and confirm non-significance.
4. Discussion: Line 259-260, complete the sentence.
5. Discussion: Is there any data on effect of DF/MP on the bone development in the developing foetus? This could open up another dimension about the effect of the NSAIDs on foetal bone development and differentiation.
6. Is it possible to use human pre-osteoblastic cell line alongside the mouse cell line? If data available please discuss.
Author Response
Please see the attachment

This manuscript is a resubmission of an earlier submission. The following is a list of the peer review reports and author responses from that submission.
Round 1
Reviewer 1 Report
In this manuscript, Lehmann et al., report in-vitro investigations in mouse calvarial preosteoblast cells (MC3T3-E1) as a follow up of their in-vivo study that showed Diclonefenac (DF) decreases trabecular bone volume and mineral density. Cell viability, caspase activity, and gene expression of osteoblast differentiation markers were examined in MC3T3-E1 cells treated with Diclonefenac (DF) and methylprednisolone (MP) to determine the in vitro effects of these NSAID/ GCs.
The methods used in the study are standard for this type of in-vitro gene expression analysis and cell proliferation. The use of two reference genes (Hprt &B2m) for the gene expression is appreciated as and is as per MIQE guidelines for RT-qPCR.
The manuscript seems to be well written. Experiments performed with necessary replicates and data analysed with appropriate statistical tools. A few suggestions listed below to improve the manuscript:
Major Comments:
1. Figure 1 : Labelling/ annotation of the treatment groups are not clearly mentioned in Figure 1 or the description. The authors have left it to the readers imagination on which of the data bars correspond to MP/DF treatment. It is therefore difficult to cross-check the results with the figure. Please add a note in the figure description on which of the bars indicate MP and DF.
2. Missing vehicle control in the experiments: DMSO was used as the solvent for MP and DF, a vehicle control (with DMSO) is essential for the experiments shown in Figure 1 & Figure 2. Also, a common vehicle control (DMSO – single concentration control) could have been used in the gene expression experiments in Figures 3 and 4. It would be great if the authors could comment on this
3. Please mention Statistical Significance Test in Figures 1 to 4 - Kindly mention the statistical analysis method used for obtaining the significance values in the legends for Figures 1, 2, 3 and 4
Minor Comments:
Methods : Section 2.4 RNA Extraction and RT-PCR is repeat of the section 2.3 : please delete the duplicated paragraph
Reference Gene name Hprt is referred to as Hart in many places
Line 161 –‘Hart and B2m were used as reference genes’ – Pls correct Hart to Hprt
Line 228 - ‘Hart and B2m were used as references for the mRNA level’ - Pls correct Hart to Hprt
Line 238 – ‘Hart and B2m were used as 238 reference genes’ - Pls correct Hart to Hprt
Line 244 – ‘Hart and B2m were used as 238 reference genes’ - Pls correct Hart to Hprt
Line 105 - Mus musculus – kindly italicize the Binomial name
Line 108 - please correct ‘100 of μg streptomycin’ to ‘100 μg streptomycin’
Line 141 – Just curious why both random hexamers and oligo(dT) primers was used for cDNA synthesis. In cDNA synthesis from mammalian RNA, only one of the two is usually used (Mostly random primers in RT-qPCR cDNA synthesis). It would be great if you can mention the concentration/ amount of oligo(dT) primer as well.
Line 158 – ‘TaqMan Fast Advanced Master mix (×2) 5 μl, TaqMan Assay ×20 0.5 μl, -It would be better to use TaqMan Fast Advanced Master mix (2x) and TaqMan Assay mix (20x)
Line 198- ‘The exact treatment time with 50 M DF decreases the viability’ - please correct to 50 µM DF
Line 197 / 208 – Could you check the statements in these sentences and make a uniform statement about for the cell viability for the 7 day MP treatment (40/70%)
Line 252- ‘Our present study shows that DF did not affect gene expression in osteoblasts MC3T3-E1’ – it would be better if you can change to : ‘ …DF did not affect gene expression of the osteogenic markers we tested’
Lines 289-290 - ‘In a study by Hadjicharalambous 289 et al., they applied 10-5 or 10-6 and 5, 10 and 15 days of treatment with DF and prednisolone’ – Request the authors to rewrite this sentence for better clarity. I presume the authors meant the concentration 10-5 or 10-6 M. Kindly use the correct form and rewrite.
Line 349 ‘were exposed to 10−6 M cortisol’ – please correct to 10−6 M
Line 349-351 –Seems a reference is missing for this sentence. Please insert the citation for the same as well as the reference in the Bibliography (Dub et al)
And lastly, if there was a mineralization assay using Alizarin red ; in addition to the cell proliferation and caspase assay on the same set of samples. If the authors could shed some light on the same during the discussion, that would be great.
Author Response
Please see the attachement

Reviewer 2 Report
The article by Lehmann et al elucidates the effect of Diclofenac and MP on the viability and the gene expression on MC3T3 cells as means to understand the mechanism of the effect of these compounds on the cell line.
Major points to be considered:
1) The introduction is insufficient to bring across why the study is being conducted. I suggest the authors describe a clear layout for the introduction points and describe each point and derive at the end why the study was conducted and what the major conclusions are. e.g. General effect of NSAIDs and GCs on bone cells in humans, mice and cell lines; effect of DF and MP on the same as in previous studies. Known mechanism of action of DF and MP, why is the mechanism unclear and why it may be different in different types of cells and its implications and lastly ending with how this study will improve understanding the mechanism and add to the existing literature.
2) In order to have a good mechanistic study, the authors should consider doing knockdown studies using siRNAs in the presence and absence of DF/MP. Dependency may be better understood by this approach. As of now, mere reduction and increase in gene levels do not give much information about the mechanism of action. If experiments are not possible at this point, the speculations in the discussion are very lengthy and can be shortened to bring out the essence of the study in succinct way. A tabular representation may help.
Other points:
1. Lines 90 and 93 say the same thing in 2 different sentences, please club.
2. Capitalise 'i' in line 98.
3. Line 105, 106 typo error.
4. Line 108, correct typo error.
5. Material and methods: sections 2.3 and section 2.4 are the same. Repeated .. please correct.
6. Line 184, change english.
7. Line 185, please clarify if it is 50 M or 50 mM? for both DF and MP.
8. In all graphs, fig. 3 and 4, please indicate in inset boxes the colors for DF and MP in the figures itself.
9. Line 203, what is PG? Improve english.
10. For not significant, write ns on the bars for all graphs. Do not leave empty.
11. 3.3 Gene expression section, mention the figures and the panel numbers/names from where the results are described.
12. Line 237: Increased intensity of grey color.
Author Response
Please see the attachement
